# Impact of Seeding and Spatial Heterogeneity on Metapopulation Disease Dynamics

Rohit Rajuladevi
Biocomplexity Institute and Initiative (UVA)
Charlottesville, United States of America

Przemyslaw Porebski
Biocomplexity Institute and Initiative (UVA)
Charlottesville, United States of America

Madhav Marathe
Biocomplexity Institute and Initiative (UVA)
Charlottesville, United States of America

Srinivasan Venkatramanan
Biocomplexity Institute and Initiative (UVA)
Charlottesville, United States of America

## ABSTRACT

Metapopulation models capture the spatial interactions in disease dynamics through mobility and mixing matrices among regions of interest. However, the impact of seeding (initialization) in such networks is not well understood. We have constructed and open-sourced metapopulation networks for countries around the world (PatchFlow) and we use them to study the effect of seeding using an extension of a discrete-time SEIR simulator, PatchSim. The impact of initialization is studied by looking at the resulting epidemic curves. We use various metrics to characterize the epidemic curves, including those based on epidemic intensity entropy. Using these, we study the impact of various levels of connectivity, skewness in seeding and spatial resolution at the national scale. We find that these effects vary across countries and are more pronounced at certain transmissibility levels. This study provides early insights into the impact of model initialization and demonstrates the use of PatchFlow networks.

## CCS CONCEPTS

• **Computing methodologies** → **Modeling methodologies**.

## KEYWORDS

Metapopulation models, Computational epidemiology

**ACM Reference Format:**
Rohit Rajuladevi, Przemyslaw Porebski, Madhav Marathe, and Srinivasan Venkatramanan. 2022. Impact of Seeding and Spatial Heterogeneity on Metapopulation Disease Dynamics. In *epiDAMIK 2022: 5th epiDAMIK ACM SIGKDD International Workshop on Epidemiology meets Data Mining and Knowledge Discovery, August 15, 2022, Washington, DC, USA*. ACM, New York, NY, USA, 4 pages.

## 1 INTRODUCTION

The COVID-19 pandemic has brought the need for diverse mathematical models for planning, response and course of action analysis. Mechanistic models based on classical theory and known as SEIR (Susceptible-Exposed-Infected-Recovered) models were used extensively throughout the COVID-19 response. Depending on the

group and use, these models ranged from simple ODE style models to more complex agent-based models. The simple models were useful and easier to calibrate but often lacked sub-national structure to effectively capture the spatial, temporal and variation in population density, mobility patterns and population demographics. Metapopulation models can capture the traditional SEIR dynamics within a network context, and can thereby represent sub-national spread. Metapopulation models provide valuable insights into individual regions as well as overall disease spread without significant computational cost. Spread of the disease within a country is dictated by many factors, such as population mixing across regions, intervention levels or vaccine uptake. Under no interventions, in a population with limited prior immunity, seeding and population structure dictate how quickly a disease can spread through the population, given disease transmissibility. Recent studies [3] have investigated this phenomenon and have observed significant differences based on the seeding conditions.

In this study, we use an open-source metapopulation simulation engine, PatchSim[6] to study the effect of seeding in metapopulation networks. In order to realistically represent the effect of population distribution and geography of various countries, we leverage an open-source dataset, PatchFlow [4], to provide PatchSim compatible metapopulation networks at admin1 and admin2 (equivalent to states and counties for the US) resolution. These networks are generated under various parameterizations of the radiation model [5]. The goal of this research is to better characterize the role of initial seeding on disease spread using various metrics, and also to demonstrate the utility of such open frameworks and datasets for pandemic preparedness. The key contributions of this paper are open-source datasets for analysis of disease dynamics by spatial seeding; the analysis of spatial distributions and their relation to temporal disease spread; and using metrics such as epidemic entropy to study the impact of seeding strategies.

## 2 EXPERIMENTAL DESIGN

The overall experiment design is shown in Figure 1. Below we briefly describe the simulation framework and the networks being used for the study.

### 2.1 PatchSim Description

PatchSim is a discrete-time metapopulation SEIR simulation engine with the ability to use various mobility and mixing networks along with fine-grained spatio-temporal control of seeding, vaccinations and interventions [6]. For the individual regions represented within

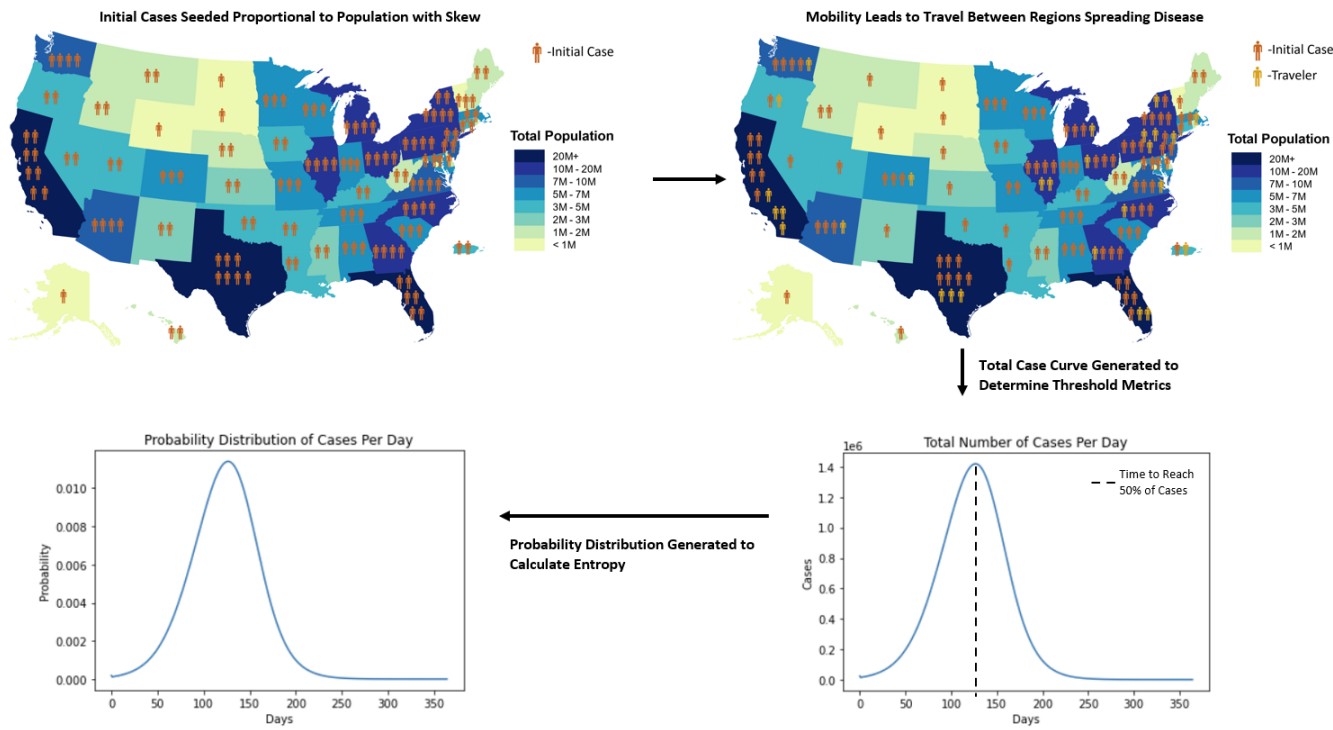

**Figure 1: Overview of experimental design**

the network, difference equations determine the disease dynamics within the region. Each step of the simulation proceeds by: (a) computing the effective population in each patch through mobility, (b) updating the disease state using proportions of the effective populations, (c) back-calculating the updated disease states for each patch. More details on the dynamics including exact equations can be found in [6, 7].

## 2.2 PatchFlow Description

PatchFlow is a set of pre-calculated networks for simulating the metapopulation SEIR model at the national scale for most countries of the world [4]. These are generated by first aligning the Gridded Populations of the World (GPWv4) [2] with the administrative region boundaries from the Database of Global Administrative Areas (GADM), by extracting the population covered by the GADM defined regions (patches) from the raster data provided by the GPWv4 . These regions at specified administrative levels were then connected using a radiation model [5]. Since the level of connectivity could vary from country to country, across spatial resolutions, as well as during different interventions (travel restrictions, lockdowns, etc.), the repository contains networks at various levels of connectivity, parameterized by the outflow parameter – the percent of the population that travels in the network. The datasets within the repository are designed to be compatible to be run with PatchSim by providing the required network and population files. Although we have PatchFlow networks at admin1 and admin2 resolutions for

220 countries, for this study we restricted to countries with a population of over 1 million. To test the effect of different characteristics incorporated into the PatchSim disease models, 4 parameters were varied: the admin level (a), radiation parameter (rad), seeding scaler (skew) and exposure rate ($\beta$). The three admin levels analyzed were admin 0, 1, and 2. Admin 0 represented a homogeneous SEIR model with one region for the entire country, which was used as comparison against the metapopulation model. The other three parameters were varied as follows: rad $\in$ {1%, 5%, 10%} representing the fraction of mobile population, skew $\in$ {1, 2, 3} used to modulate the seeding across patches, $\beta \in$ {0.4, 0.45, 0.5, 0.55, 0.6, 0.65, 0.7} representing the transmissibility. Infectious and incubation periods were fixed throughout the simulation. The total number of initial cases $S$ was set as a proportion of the national population (P), in this study 10 per 100,000 people (i.e., $S = 0.01\%P$). Initial cases within the regions were allocated as a proportion relative to the patch's population (p) exponentiated by the skew parameter:

$$InitialCases = \frac{p^{skew}}{\sum p^{skew}} \cdot S \qquad (1)$$

As seen in Figure 2 higher skew parameter lead to more initial cases concentrated in higher populated patches as shown below (blue:skew=1, orange:skew=2, green:skew=3).

Generated from the parameter variation was a dataset containing the parameter values of the simulation and metrics of the disease curve for each country.

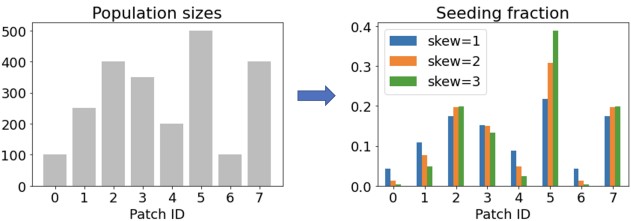

**Figure 2: Seeding proportion based on population size and skew parameter**

## 2.3 Metrics

To evaluate the effect of the changed parameters we used the number of days to cross 1%, 50%, and 99% of total cases and the Shannon Entropy measurements of the normalized epidemic curve. The epidemic entropy measurement was calculated as follows:

$$H(X) = -\sum_{t=1}^{n} P(x_t)log(P(x_t)) \tag{2}$$

where $P(x_t) = \frac{I(t)}{C_T}$ is the fraction of cases on day $t$ of the epidemic, with $I_t$ the incident epidemic curve, and $C_T$ the cumulative cases by the end of simulation on day $T$. This metric has been used in the past to characterize the intensity of seasonal influenza epidemics and the role of urban-ness and humidity in shaping the disease dynamics across various cities [1]. In addition to the epidemic entropy, the population entropy of an admin level was calculated to provide context to the population distribution of the admin level. Using the same equation for entropy, $P(x_i) = \frac{p_i}{P}$, the fraction of the population of a region to the total population.

## 3 RESULTS

Different disease curves resulting from the parameter variation illustrated features unique to a metapopulation model. Regions and seeding strategies that would have had similar epidemic characteristics under a simple SEIR model instead produced distinct curves with characteristics only present in a metapopulation model.

### 3.1 Relationship between spatial and temporal entropy

The epidemic entropy was analyzed in conjunction with the population entropy to characterize patterns emerging from the spatial distribution of a population. Displayed in Figure 3 are the scatter plots of the population entropy values of each country for selected parameters shown in relation to the corresponding epidemic entropy for the country. Shown is a positive relationship between the population entropy and the epidemic entropy which becomes more positive with increases in the skew parameter. This indicates that higher spatial entropy tends to lead to higher temporal entropy. This could be due to cases spreading in more populated areas earlier with higher skew values. In countries with large differences in population by region and thus low population entropy values, skew leads to rapid disease spread in the highest populated regions which could lead to a sharper peak and lower epidemic entropy. Otherwise for countries with uniform populations and thus higher

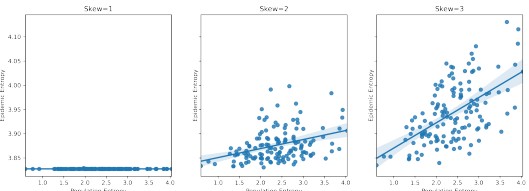

**Figure 3: Epidemic entropy and population entropy regression plot. .**

population entropy values, the disease will need to spread to many regions to infect a sizable percentage of the population. From this, seeding more cases in slightly more populated regions leads to a longer time to peak and thus a higher epidemic entropy value.

These trends between spatial distribution and epidemic entropy can be analyzed at a country level. Displayed in Figure 4 is a heat map comparing entropy values between Mongolia, Slovakia, and the United States of America, each with different population sizes, density and distribution. Slovakia has a fairly uniform population distribution at both the admin 1 level and admin 2 level. Meanwhile, Mongolia has an uneven distribution. Nearly 50 percent of Mongolia's population lives in the capital city of Ulaanbaatar. It can be

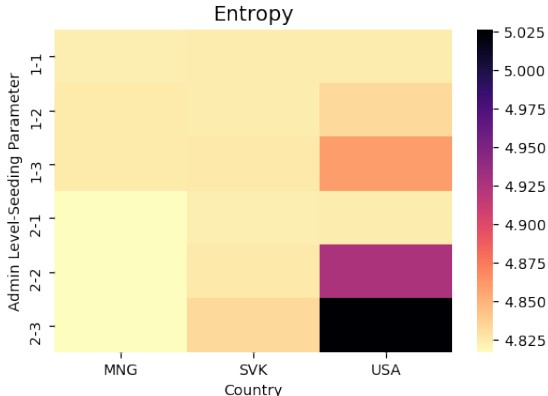

**Figure 4: Entropy values for Mongolia, Slovakia, and the United States of America.**

seen that a higher skew seemed to contribute to higher epidemic entropy values, since this led to more concentrated seeding, and hence longer for the epidemic to reach all patches. The United States and Slovakia had the greatest epidemic entropy value when in admin 2 with a skew value of 3 while Mongolia had its greatest when in admin 1 with a skew value of 3. Compared to Mongolia and Slovakia, the absolute increase in epidemic entropy was much more apparent for the United States of America. Slovakia and Mongolia on the other hand had only slight deviations in epidemic entropy. Slovakia seemed to have a greater epidemic entropy change when the skew was varied while Mongolia had a greater change with variation in the admin level. Slovakia's epidemic entropy measurements seemed to be more dependent on the skewing parameter because it

resulted in a disproportional initial case distribution. Meanwhile, Mongolia's epidemic entropy seemed to be more dependent on the admin level because the population distribution was already so skewed. Therefore, regardless of the skewing value, Ulaanbaatar always represented a high proportion of initial cases.

## 3.2 Entropy and timings variations for a given country

Figure 5 displays the variation of the disease spread parameters and the resulting epidemic entropy values for the United States. In

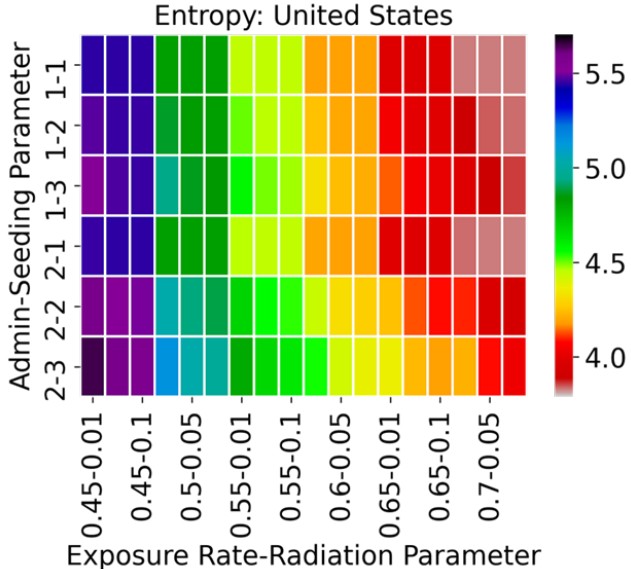

**Figure 5: Entropy heat map of the United States.**

the range of $\beta$ explored, the impact of transmissibility on epidemic entropy is monotonic, i.e., a higher $\beta$ led to sharper epidemics, and thus lower epidemic entropy. However, it is to be noted that, for much lower $\beta$, it could lead to faster extinction of the epidemic process (and hence also lower epidemic entropy). Thus, in general, the impact of transmissibility can be non-monotonic. With regards to admin level, with a fixed $\beta$, as admin level increased, epidemic entropy increased due to localized mixing within the regions, and fewer long-range connections. This was further accentuated by the skew in seeding and the rad parameter of the network. Finally, uniform seeding (skew=1) did not significantly change peak times, whereas for higher skew, the effect of the rad parameter was more pronounced. This effect is better visualized in Figure 6. In admin 1, increasing the skew and varying the radiation parameter did not have as strong effect on the peak timing. This is likely due to more comparable population sizes at admin1 resolution than at admin2 resolution.

## 4 CONCLUSIONS AND FUTURE WORKS

We have used open-source framework and datasets for pilot studies on the impact of seeding on disease dynamics over metapopulation networks. We note that the seeding impact results from the

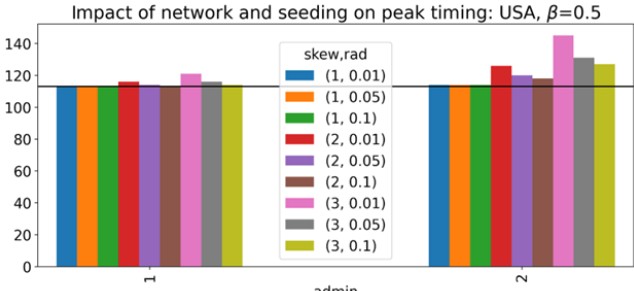

**Figure 6: Peak timing of cases varying skewing and radiation parameter per admin resolution.**

subnational population distribution, as demonstrated by the correlation of population entropy to the epidemic entropy. The study also highlights the role played by the level of connectivity between the regions. While we observe general trends across parameters, we also note that these trends may not be universal, and could vary based on country-specific conditions. The seeding strategy can be expanded to test the impact of central or peripheral node seeding as done in [3] for real-world metapopulation models. While we have begun with seeding as a case study, one can observe a variety of dynamics when interventions[8] and vaccinations [9] are introduced in a spatially heterogeneous fashion. Future work will focus on improving understanding of such modeling frameworks for better pandemic preparedness.

## ACKNOWLEDGMENTS

This work was supported by NSF RAPID 2027541: COVID-19 Response Support: Building Synthetic Multi-scale Networks, NSF Expeditions 1918656: Collaborative Research: Global Pervasive Computational Epidemiology, DTRA subcontract/ARA S-D00189-15-TO-01-UVA.

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
