# OpenReview forum: "Impact of Seeding and Spatial Heterogeneity on Metapopulation Disease Dynamics"
_ACM.org/SIGKDD/2022/Workshop/epiDAMIK — KDD 2022 Workshop epiDAMIK Oral_

### Official Review · Reviewer_YCa3 · 2022-06-19
**The paper is clearly written and easy to understand. However, the contribution is limited.**

**Rating:** 4
**Confidence:** 3

**Review:**

The paper studies an interesting problem of investigating the effect of seeding in the simulations. Here is a list of strengths, weaknesses, and suggestions of the paper

- Strengths

The paper is clearly written, and the result of varying the seeds based on the skewness parameter is promising. The visualizations help what the authors are conveying in the paper.

- Weaknesses

The contribution is limited - their work studies the effect of seeding but just explores the space of the skewness. Furthermore, the authors do not justify the choice of the parameters, e.g., S = 0.01%P, skew in {1,2,3}, beta range, and the metric of crossing 1%, 50%, 99%.

- Suggestions

I suggest the authors explore some other seeding methods, such as (i) selecting seeds randomly, (ii) based on centrality measures, e.g., high degree, low degree, betweenness, etc. Also, I suggest authors provide justification or explore a wider range of the parameters (sweep through the space). I suggest authors tune the simulation parameters to some specific infectious diseases based on the literature, and give a set of results per disease. Some other minor suggestion is to add some context to the figure captions.

---

### Official Review · Reviewer_Hajp · 2022-06-27
**This paper focuses on using the metapopulation simulator and SEIR model to evaluate the impact of seeding and spatial heterogeneity on COVID spread in the metapopulation network.**

**Rating:** 4
**Confidence:** 4

**Review:**

This paper focuses on using the metapopulation simulator and SEIR model to evaluate the impact of seeding and spatial heterogeneity on COVID spread in the metapopulation network.

Pros
1. Multiple groups of the parameter ($\beta$, $skew$) are explored for evaluation.
2. The explanation that higher spatial entropy leads to higher temporal entropy makes sense.

Cons:
1. More discussion on the SEIR model can be added.
2. If possible, maybe a set of calibrated parameters on real-world data can also be used for evaluation.

---

### Official Review · Reviewer_GJGH · 2022-06-27
**Epidemic intensity and meta population: Good paper, presentation can be improved**

**Rating:** 4
**Confidence:** 5

**Review:**

The paper studies the impact of seeding on the epidemic intensity. A simulation engine called PatchSim is used to simulate a meta-population SEIR process with various mobility, mixing, and transmissibility parameters.

Pros:
- Analysis along interesting dimensions of spatial heterogeneity and epidemic intensity
-
Cons:
- Contributions could be better clarified
- Some more metrics could be used for assessing the epidemic impact
- Presentation issues

See below for detailed comments:

- The statement of contributions or findings could be more specific. Currently, the abstract and introduction present it at a high level. Some specific results could be highlighted. This will help assess the contributions and strengthen the work.
- While epidemic entropy is a useful metric, it may not convey the full picture. Small random "peaks" and large multiple peaks would be normalized to the same entropy value. So, perhaps, the intensity of the peaks could also be discussed. Some epidemic curves could also be presented from the simulations with corresponding entropies to help the readers.
- Presentation issues
  - Figure 4: Is the entropy difference significant (4.825 - 5.025)? Perhaps the authors can provide some justification in the paper. Also some epidemic curves highlighting the difference would be convincing.
  - Figures labels can be improved to better convey the information: e.g., (1) Figure 3, y-axis label ("Epidemic Entropy") (2) Figure 4 y-axis label is confusing. Could be replaced with "(Admin-level, skewness)"
  - References [5] and [6] are identical.